# Comparison of Velocity and Estimated One Repetition Maximum Measured with Different Measuring Tools in Bench Presses and Squats

**DOI:** 10.3390/s24237422

**Published:** 2024-11-21

**Authors:** Roland van den Tillaar, Hallvard Nygaard Falch, Stian Larsen

**Affiliations:** Department of Sport Sciences and Physical Education, Nord University, 7600 Levanger, Norway; falch7@hotmail.com (H.N.F.); stian.larsen@student.nord.no (S.L.)

**Keywords:** peak velocity, mean velocity, mean propulsive velocity, resistance training, strength, 1-RM

## Abstract

The aim of this study was to compare barbell velocities at different intensities and estimated 1-RM with actual 1-RM measured with different measuring tools in bench presses and squats. Fourteen resistance-trained athletes (eight men, six women, age 28.1 ± 7.5 years, body mass 78.1 ± 12.2 kg, body height 1.73 ± 0.09 m) performed bench presses and squats at five loads varying from 45 to 85% of one repetition maximum (1-RM), together with 1-RM testing, while measuring mean, mean propulsive, and peak barbell velocity with six different commercially used inertial measurement units (IMUs) and linear encoder software systems attached to the barbell. The 1-RM was also estimated based upon the load–velocity regression, which was compared with the actual 1-RM in the bench press and squat exercises. The main findings were that GymAware revealed the highest reliability along with minimal bias, while Musclelab and Vmaxpro showed moderate reliability with some variability at higher loads. Speed4lifts and PUSH band indicated greater variability, specifically at higher intensities. Furthermore, in relation to the second aim of the study, significant discrepancies were found between actual and estimated 1-RM values, with Speed4lifts and Musclelab notably underestimating 1-RM. These findings underscore the importance of selecting reliable tools for accurate velocity-based training and load prescription.

## 1. Introduction

When training for increasing strength, muscle hypertrophy, and power output for the upper and lower body, both bench presses and squats are popular resistance training exercises [1,2]. Training load or proximity-to-failure may be a crucial training variable for different physiological adaptations (i.e., strength and muscle hypertrophy) following resistance training [3]. Unlike traditional one-repetition maximum (1-RM) testing, which may require the lifter to perform repetitive maximal lifts to track 1-RM progression, estimating the load–velocity relationship through a single 1-RM test could enable continuous tracking of the load–velocity relationship, and thus continuous tracking of changes in the lifters’ performance [4]. Hence, velocity-based resistance training has become popular as this could measure concentric velocity and therefore prescribe training load [5]. The use of concentric velocity to prescribe load is justified by the linear relationship between velocity and the % of 1-RM used [5,6]. Furthermore, decreased velocity or greater velocity loss indicates a decreased proximity-to-failure and increased neuromuscular fatigue [7].

It is acknowledged that three-dimensional (3D) motion capture systems are the gold standard for measuring barbell velocity [6]. However, as the price of such motion capture systems is often expensive, the proliferation of several available devices, such as inertial measurement units (IMUs), linear position transducers, linear velocity transducers, and smartphone applications have become popular to measure concentric velocity among trainers and practitioners [4,5,8]. Importantly, divergent findings related to both validity and reliability to measure mean velocity, mean propulsive velocity, and estimated 1-RM has been reported in the literature [6,7,9,10,11,12,13,14].

For example, Bosquet, et al. [10] assessed both the validity and accuracy of using a Musclelab linear encoder to estimate 1-RM in bench presses from the load–velocity relationship. This is the relationship between the load lifted and the corresponding barbell velocity at which it is lifted during different loads on which, through a linear regression analysis, a relationship is found that is often used to predict 1-RM loads in different exercises [15]. Bosquet, et al. [10] reported high correlations (r = 0.93) between the estimated and actual 1-RM. However, on average, the Musclelab software significantly underestimated 1-RM with 5.4 kg (bias). Therefore, according to the authors, this estimation is not accurate enough to prescribe resistance training intensities based upon the 1-RM estimations predicted by that software [10]. Importantly, other measurement tools for measuring velocity exist. Martínez-Cava, et al. [12] compared both inter- and intra-device agreement in squats and bench presses between T-Force, Speed4lifts, My Lift, and 3D motion capture systems. Acceptable reliability criteria were set to the smallest detectable change of <0.07 m/s. The authors concluded that the T-Force device was the preferrable option to identify potential technical errors among monitoring technologies due to the high observed intra-device reliability across the load–velocity spectrum, where the authors observed the smallest detectable change to be 0.01–0.02 m/s. In addition, they concluded that Speed4Lifts and the 3D motion capture system were fine alternatives to T-Force when training with medium and high loads, as the smallest detectable changes were 0.01–0.05 m/s and 0.02–0.04 m/s for the Speed4lifts and 3D motion capture system, respectively. In another study, both PUSH band 2.0 and Speed4lifts were checked for test-retest reliability against a 3D motion capture system in both squat and bench press exercises [7]. The authors found that both devices were reliable with slow (<0.65 m/s) and moderate (0.65–1.00 m/s), but not fast (>1.0 m/s), velocities, as they observed intraclass correlations of >0.7 for both devices during the slow and moderate velocities and intraclass correlations of <0.64 for the fast barbell loads. Complementarily, Feuerbacher, et al. [9] assessed validity and intraday reliability between the IMU-based Vmaxpro to a Vicon 3D motion capture system and a T-Force sensor during a progressive squat 1-RM test. The researchers observed that Vmaxpro had high correlations (R^2^ > 0.93) when compared with both the motion capture system and T-Force. However, Vmaxpro overestimated mean velocities with an average of 0.06 m/s compared with the other velocity measurements with large limits of agreements. Furthermore, the observed systematic bias with the other two measurement tools increased with higher mean velocities, indicating a heteroscedasticity when measuring mean velocity with different loads. Additionally, GymAware is commonly used in the scientific literature to measure velocity [16]. Recently, Janicijevic, et al. [16] compared the reliability of mean concentric velocity and peak velocity between GymAware and T-Force. The authors reported no significant differences between the devices and concluded that that both devices could be used as gold standards in research studies designed to validate other devices.

However, to the best of our knowledge, no studies have assessed reliability and validity between the velocity measurements in T-Force compared with Speed4lifts, Vmax, Musclelab, PUSH band, and GymAware with the use of mean propulsive velocity, mean velocity, and peak velocity in both the bench press and back squat exercise in the same study. As many training studies use one of these sensors to measure velocity at different loads, it could be of importance to know if these measurements with different loads are comparable between these measurement tools. This makes it easier for trainers and researchers to compare findings performed with these different measurement tools with each other and if they need to adjust results when comparing these measurement tools. Also, to the best of our knowledge, no studies have compared the estimated 1-RM with the actual 1-RM with the use of both mean propulsive and mean velocity for these velocity measurement devices. These are the parameters that are often used to estimate 1-RM; therefore, it is not known how accurate the use of these parameters are. Therefore, the first aim of this study was to assess reliability and validity when measuring mean propulsive, mean, and peak velocity between T-Force, which is considered as the gold standard, to five other velocity measurement devices at 45%, 55%, 65%, 75%, 85%, and 100% of 1-RM in the squat and bench press. This range of loads was used as strength training occurs in this whole range with these measurement tools. The second aim of this study was to compare actual 1-RM with estimated 1-RM and assess reliability for these estimations with the use of mean propulsive velocity and mean velocity for six different velocity measurement devices with the use of linear regression. It was hypothesized that T-Force and GymAware would have the least mean bias and narrowest limits of agreement, as these measurements often are considered the gold standard [16].

## 2. Materials and Methods

To compare the six different measuring tools, a within subject design was used in which all measuring tools were attached while performing the squat and bench press exercises at different intensities. Mean propulsive [17], mean, and peak velocities were the dependent variables tested across five different loads, and they were used to investigate the impact of load and exercise type on the outcome measures together with 1-RM load testing. Participants completed two repetitions at each load at low loads (≤65% of 1-RM) and one repetition at higher loads.

### 2.1. Subjects

To determine adequate sample size between estimated and actual 1-RM measurements between the six measurement tools, an a priori power analysis was calculated using G*Power (version 3.1.9.2, University of Kiel, Kiel, Germany) using the f test family (ANOVA repeated measures, within factors, f = 0.8, α = 0.05, and power of 0.80) and based upon the findings of Bosquet, et al. [10] and Fitas, et al. [18]. The analysis revealed that a total sample size of n = 12 would be sufficient to find significant and medium-sized effects of estimated 1-RMs of measurement tools with actual 1-RM loads. Thereby, 14 resistance-trained athletes (8 men, 6 women, age 28.1 ± 7.5 years, body mass 78.1 ± 12.2 kg, body height 1.73 ± 0.09 m) with a minimum of two years of at least two times per week of resistance training experience participated in the study in which they trained squats and bench press. Inclusion criteria specified that subjects had to manage a squat equivalent to 1.5 × body mass (men) and 1 × body mass (women) [19], following the technique requirements established by the International Powerlifting Federation. For bench presses, men had to lift in 1-RM at least 1.2 × body mass and women had to lift 0.9 × body mass [20] to ensure that they have an appropriate lifting technique in both exercises. Additionally, subjects had to declare the absence of any injury or illness, which could hinder maximum effort. The participants were instructed to avoid undertaking any resistance training during the 48 h prior to testing. Each participant was informed of the testing procedures and possible risks, and written consent was obtained prior to the study. The study complied with the current ethical regulations for research, was approved by the Norwegian Center for Research Data (project number: 991974), and conformed to the latest revision of the Declaration of Helsinki.

### 2.2. Procedures

Participants were randomized using an online randomizer to start with either the bench press or squat to avoid the exercise order influencing the results. After randomization, a specific warm-up for the given exercise was initiated. Subjects began by performing the exercise with only the barbell (15 kg for females, 20 kg for males) as external resistance, with instructions to perform each repetition with maximal velocity. The first set consisted of 5–10 repetitions to allow participants to get accustomed to the equipment. Subsequent sets were performed with 2 repetitions in random order at 45%, 55%, 65%, and 75% and one repetition at 85% of estimated 1-RM. Afterwards, 1-RM attempts were performed based on prior self-reported estimations, with each successful lift leading to a 2.5% increase in load until the actual 1-RM was established. The estimated 1-RM in each exercise was based upon 1-RM testing 2–3 weeks prior to the current test. Actual 1-RM was determined for all subjects in fewer than 3 attempts. After establishing the 1-RM for the first exercise, the same procedure was repeated for the second exercise. A minimum of 3 min of rest was included between each set and a minimum of 5 min between exercises was included to reduce the influence of fatigue. The subjects performed the bench press and deep squat according to the rules and regulations set by the International Powerlifting Federation. Thus, the depth through the end of the descent phase during the squat was measured and standardized with the depth requirement from the International Powerlifting Federation and marked with a horizontal band behind the subject, which needed to be touched to start the ascent. For the bench press, the requirement for a full stop on the chest was removed; they were allowed to touch and press, but no bounce of the barbell was allowed. Both stance width, barbell placement, and external rotation of the feet during squats and grip width during the bench press were self-selected by the subjects but standardized for each individual throughout all loads. The descent movement velocity was self-selected, but the ascent phase had to be performed with maximal intensity in each attempt.

### 2.3. Measurements

The six different measuring systems (four linear encoders and two IMU-based systems) were all attached to the barbell. The two IMU-based systems were PUSH band 2.0 (PUSH Inc., Toronto, ON, Canada) and Vmaxpro (Blaumann & Meyer—Sports Technology UG, Magdeburg, Germany). Prior to the test, PUSH band and Vmaxpro were attached to the barbell according to the specifications of the manufacturer. PUSH band 2.0 and Vmaxpro consists of a 3-axis accelerometer and a gyroscope that provides 6 degrees of freedom in its coordinate system sampling at 200 Hz. PUSH band and Vmaxpro were validated before by Callaghan, et al. [7] and Feuerbacher, et al. [9], respectively. To process PUSH band data, a Butterworth filter was used to smooth the acceleration data, and vertical velocity was calculated by the integration of the vertical acceleration with respect to time. Readers are referred to Balsalobre-Fernandez, et al. [21] for further information on the specific calculation methods. PUSH band displayed mean velocity during the upwards phase of the exercises. PUSH band was linked to an iPad Air (model A1474, Apple Inc., Cupertino, CA, USA) with the PUSH application (version 4.6.2, PUSH Inc., Toronto, ON, Canada) via Bluetooth to record the measured data. Instant velocities were obtained, and mean and peak velocities were calculated for the upward phase and visualized using the Vmaxpro application (version 4.2), which was connected to an IOS device (IPhone 11; Apple, Inc., Cupertino, CA, USA) via a Bluetooth 5.0 connection.

All the four linear encoders were attached to the barbell (two on the left and two on the right side) according to the recommendations of the manufacturers. Speed4lifts (Speed4lift, Madrid, Spain) was validated by Callaghan, et al. [7] and Martínez-Cava, et al. [12] and sampled at 100 Hz, connected with an iPad Air with the Speed4lifts application (version 2.4.7, Speed4lifts, Madrid, Spain) via Wi-Fi. The mean propulsive and peak velocity of each repetition recorded by Speed4lifts was later exported for analysis. Mean velocity is the average velocity of the entire upward phase of the lift, while mean propulsive velocity is the average velocity during the propulsive phase, defined as the portion of the upward phase where acceleration is greater than gravity [17].

GymAware (Kinetic Performance Technologies, Canberra, Australia) has been validated by Orange, et al. [13], who measured the vertical displacement of the barbell from the rotational movement of the spool, in which displacement data were time-stamped at 20 ms time points to obtain a displacement–time curve for each repetition, which was down-sampled to 50 Hz for analysis. Instantaneous velocity was determined as the change in barbell position with respect to time and mean and peak velocity of the upward phase and was calculated by the software. Data obtained from GymAware were transmitted through Bluetooth to a tablet (iPad; Apple, Inc., Cupertino, CA, USA) using the GymAware v2.1.1 app.

The T-Force system (Ergotech, Murcia, Spain) validated by Garnacho-Castaño, et al. [11] is an isoinertial dynamometer that consists of a cable-extension linear velocity transducer interfaced with a personal computer by means of a 14-bit resolution analog-to-digital data acquisition board. Instantaneous velocity was automatically calculated at a sampling rate of 1000 Hz by the custom software v.2.28. Mean propulsive, mean, and peak velocity for the upward phase were calculated and used for further analysis.

The Musclelab (Ergotest Technology AS, Langesund, Norway) encoder validated by Bosquet, et al. [10] measured the upward phase duration of the barbell with a resolution of 0.075 mm with a 200 Hz sampling rate. Mean propulsive, mean, and peak velocity of the barbell of the upward phase was calculated by using a 5-point differential filter with Musclelab v10.73 software (Ergotest Technology AS, Langesund, Norway). The linear encoder was connected via a data synchronization unit to a laptop.

Two devices (T-Force and Musclelab) reported all three parameters, while five reported mean (T-Force and Musclelab, GymAware, Vmaxpro, PUSH) and peak velocity (Speed4lift, T-Force and Musclelab, GymAware, Vmaxpro) and three reported mean propulsive velocity (Speed4lift, T-Force, and Musclelab). The highest velocity of the two repetitions at the lower loads (45–75% of 1-RM) and velocity of the higher load (85% to 1-RM) from each system was used for comparison between the systems at each load for each exercise. Because only three systems reported mean propulsive velocity, this propulsive velocity was also compared with the mean velocities of these systems and other systems.

The estimate of 1-RM mean propulsive and/or mean velocities from each device at each of the different loads (45–85% of 1-RM) were used to establish a load–velocity relationship as a product of the load and velocity (five points) for each participant. Based on the athlete’s performance with the various loads, a linear regression was applied to calculate the theoretical 1-RM for each participant.
*y* = *a* ∗ x + *b*

Both the coefficient of x (*a*) and the y-intercept (*b*) is individual for each subject. To establish *a* and *b* in the linear equation for each participant, a scatter plot with an added linear regression line was calculated in Excel. By replacing x (*a*) with 0.18 m/s [22] in the formula for bench press and 0.32 m/s for squats [14], the load–velocity relationship for maximal performance was established for each participant, and the theoretical 1-RM could be calculated. This theoretical 1-RM was compared with the actual 1-RM in both exercises.

### 2.4. Statistical Analyses

Normality was checked with the Shapiro–Wilk test. Reliability was assessed through the coefficient of variation (CV) and intraclass correlation coefficient (ICC) between the T-Force system and the other velocity devices. T-Force was used as it has been considered the gold standard in the absence of 3D motion capture systems [16,23] and is often used to validate and test new velocity devices [24,25]. Acceptable reliability was considered as an ICC > 0.70 and a CV < 10% [23,24]. The mean bias and 95% limits of agreement (LOAs) were calculated with regards to the method developed by Bland and Altman [26] with one-way ANOVAs performed on the bias between T-Force with the other measurement tools and between the estimated and actual 1-RM loads with repeated measurements for the squat and bench press. Pearson’s product moment correlation was used for evaluating the association between estimated 1-RM and actual 1-RM. A correlation between 0.5 and 0.69 was considered moderate, between 0.70 and 0.89 was considered high, and over 0.9 was considered very high [27]. To compare actual 1-RM and estimated 1-RM measured with the different devices at different loads, a one-way ANOVA (estimated 1-RM and 1-RM) with repeated measurement was used. Where the sphericity assumption was violated, the Greenhouse–Geisser-corrected *p*-values in the results were reported. Post hoc tests using the Holm–Bonferroni probability adjustment were used to identify differences. The effect size used and reported in this study was partial eta squared (η^2^), where 0.01 ≤ η^2^ < 0.06 constituted a small effect, 0.06 ≤ η^2^ < 0.14 constituted a medium effect, and η^2^ < 0.14 constituted a large effect [28]. The level of significance was set at *p* ≤ 0.05 for all tests, and the analyses were carried out with JASP v0.17.3 (University of Amsterdam, Amsterdam, The Netherlands).

## 3. Results

### Reliability Velocity Measures

Velocity values only reached acceptable levels across all loads for GymAware peak velocity compared with the T-Force device during both the squat and bench press (Table 1 and Table 2). Moreover, Musclelab achieved acceptable reliability for all loads except for 100% load for mean propulsive velocity, mean velocity, and peak velocity in both the squat and bench press exercises, respectively. Also, Speed4Lifts achieved acceptable reliability except at 85% load for mean propulsive velocity and peak velocity in the squat. For the bench press, Speed4Lifts did not reach acceptable reliability for 100% load for mean propulsive velocity and all loads except for 65% for peak velocity. Vmaxpro reached acceptable reliability for all loads except for 85% and 100% for mean velocity and 100% load for peak velocity in the squat; meanwhile, the GymAware device did not reach acceptable reliability for 100% barbell load for either mean velocity or peak velocity in the bench press but reached acceptable reliability for all other loads. Finally, PUSH band reached acceptable reliability for all loads except the 100% load in the squat, whereas for the bench press, acceptable reliability was only achieved for the 45 and 65% loads (Table 2).

A significant bias was found as shown by the Bland–Altman plots and one-way ANOVAs between the T-Force with the other measurement tools in squat (F ≥ 12.8, *p* < 0.001, η^2^ ≥ 0.19, Figure 1) and bench press (F ≥ 3.37, *p* ≤ 0.037, η^2^ ≥ 0.05, Figure 2) exercises for the propulsive, mean, and peak velocities. No significant heteroscedasticity was found as indicated by the linear regressions between the T-Force and the other measuring tools for any of the parameters (r ≤ 0.34, *p* > 0.05, Figure 1 and Figure 2). A post hoc comparison revealed that for squats, Speed4lift underestimates propulsive and peak velocity compared with T-Force, while Musclelab significantly overestimates in all variables. PUSH, Vmaxpro, and GymAware overestimate only in mean velocity (Figure 1). For bench presses, Musclelab significantly overestimates mean propulsive and mean velocity, PUSH and Vmaxpro overestimate in mean velocity, and GymAware and Speed4lift underestimate in peak velocity (Figure 2).

When performing a one-way ANOVA of repeated measurements per load, a significant effect between equipment was found for mean propulsive velocity at loads 45–75% of 1-RM (F ≥ 5.9, *p* ≤ 0.01, η^2^ ≥ 0.37), for mean velocity at loads 55–100% of 1-RM (F ≥ 6.8, *p* ≤ 0.001, η^2^ ≥ 0.58), and for peak velocity at 45 and 55% of 1-RM loads (F ≥ 3.0, *p* ≤ 0.030, η^2^ ≥ 0.23). A post hoc comparison revealed that only Musclelab recorded significantly higher mean propulsive velocity than T-Force at 45% of 1-RM. PUSH band recorded significant higher mean velocities than the other tools at all loads except for 45% of 1-RM, while GymAware recorded significantly higher velocities at loads of 55–75% of 1-RM and Vmaxpro at 55 and 75% of 1-RM than T-Force. For peak velocity, significant differences were only found at lower loads (45 and 55%) with higher velocity with Musclelab and lower velocities in Speed4lifts compared with T-Force (Figure 3).

For bench presses, a significant effect between equipment was found for mean velocity at loads 45–65% of 1-RM (F ≥ 3.56, *p* ≤ 0.014, η^2^ ≥ 0.26) and for peak velocity at loads 45, 65, and 75% of 1-RM (F ≥ 2.59, *p* ≤ 0.048, η^2^ ≥ 0.17). However, a post hoc comparison revealed that in mean velocity, PUSH band recorded significantly higher velocities than T-Force at load 45–65% of 1-RM in addition to significantly higher mean velocities of Vmaxpro with T-Force at 65% of 1-RM. For peak velocity, significant differences were only found at 65% of 1-RM with higher velocity with Vmaxpro compared with T-Force (Figure 4).

A significant effect of measuring equipment was found when mean propulsive velocity was used to estimate 1-RM compared with actual 1-RM in both bench press and squat exercises (F ≥ 5.03, *p* ≤ 0.005, η^2^ ≥ 0.30). However, when using mean velocity, no significant differences were found between actual and predicted 1-RM in both exercises (F ≤ 1.37, *p* ≥ 0.25, η^2^ ≤ 0.12). A post hoc comparison revealed that Speed4lift, when using mean propulsive velocity as the calculation, underestimated 1-RM significantly in squat (−5.8%) and bench press (−9.2%), together with Musclelab underestimating 1-RM in bench press (−4.8%) (Table 3 and Table 4).

## 4. Discussion

The main objective of the current study was to assess the reliability of measuring mean propulsive, mean, and peak velocities between the T-Force device and five other velocity measurement devices (GymAware, Speed4lifts, Musclelab, Vmaxpro, and PUSH band) at varying intensities of 1-RM in both squats and bench presses. The second objective was to compare actual versus estimated 1-RM across various velocity measurement tools.

### 4.1. Reliability of Measuring Mean Propulsive, Mean, and Peak Velocities

When comparing with the T-Force over all loads during squat and bench press, almost all other measurement tools had an overestimation of the mean propulsive, mean, and peak velocity, except Speed4lift and GymAware had an underestimation at peak velocity during bench press (Figure 1 and Figure 2). However, when specified per load only, PUSH 2.0 during squats at most loads overestimated mean velocity, together with Vmaxpro and GymAware at loads varying from 55 to 75% of 1-RM. During bench press, mean velocity was only overestimated with PUSH 2.0 at loads 45–65% and Vmaxpro at 65% of 1-RM. This indicates that at lower loads, resulting in higher velocities, both the accelerometer- and encoder-based tools may face challenges. Accelerometer-based tools like Vmaxpro and PUSH band are prone to cumulative errors, which may occur due to the integration of acceleration data compounded by noise as a result of sensitivity to placement. Furthermore, the most likely explanation for the encoder-based tools is limitations in sampling rates, reducing precision. However, both acceleration- and encoder-based tools might experience issues in data smoothing. Like in squats, more significant differences in mean velocity at different loads were found between the different measuring tools compared with bench presses (Figure 3 and Figure 4), which can be attributed to the fact that during squats, the bar path is much longer than in bench presses. Thereby, more datapoints are used to calculate mean velocity more accurately, causing significant differences between the tools.

Yet, when analyzing CV and ICCs between the T-Force and other measuring tools, the CV increased and ICC decreased in general at the highest loads (85% and 1-RM), indicating that much variability occurs in the measurements at these loads. A reason for this finding can be greater technical variability as a result of fatigue at high loads, whereby the participant self-organizes to reduce the requirements of certain muscles to further overcome the external load [29].

When evaluating the measuring tools per tool, GymAware demonstrated high reliability and minimal bias across various measurements, with mean biases ranging from −0.023 to −0.007 and narrow limits of agreement indicating consistent accuracy in both squats and bench presses for mean propulsive velocity, mean velocity, and peak velocity, with main bias and LOA values consistently being within small ranges. The intraclass correlation coefficients for GymAware were also very high across most measurements (ICC = 0.72 to 0.96) but did not reach acceptable reliability for 100% barbell load for either mean velocity (ICC = 0.60 at 100% load) or peak velocity (ICC = 0.79 at 100% load) in the bench press, indicating reduced precision at high loads. The high precision of GymAware was unsurprising, being referred to as “the gold standard” [16], as previous research has emphasized the importance of high sampling rates and linear encoders for accurate measurements [11]. Furthermore, the findings of our study are supported by Orange, et al. [13], who used GymAware and observed good-to-excellent reliability in both the bench press and squat for 40–90% of 1-RM; however, they did not test 1-RM. Musclelab also revealed reliable performances in all loads except not at 1-RM loads, where the ICC of mean velocity in bench press was too low (ICC = 0.65) and the CVs for all velocity parameters at this load in both exercises were too high (CV ≥ 12.3, Table 1 and Table 2). However, a systematic mean bias of higher measured velocities over all loads was found for almost all parameters in both exercises (Figure 1 and Figure 2), indicating that this measurement system in general measures higher velocities. This is also visible in the mean propulsive velocity at 45% of 1-RM loads during squats, where the measured velocity was significantly higher than with the T-Force (Figure 3). Yet, even with the systematic bias, the LOAs varying from 0.0775 (mean propulsive velocity in bench press and mean velocity in squats) to 0.115 (peak velocity in squats) were the lowest of the measurement tools in relation to the T-Force, indicating consistency (Figure 1 and Figure 2).

The Vmaxpro demonstrated similar reliabilities to Musclelab with an also too low ICC of 0.69 with peak velocity during squats and too low CVs with maximal loads (Table 1 and Table 2). Furthermore, Vmaxpro also measures higher mean velocities compared with the T-Force, as shown by significant mean biases (Figure 1 and Figure 2) and higher velocities at loads of 55 and 75% in squats (Figure 3) and 65% in bench presses (Figure 4). This was in line with Feuerbacher, et al. [9], who also observed an overestimation of mean velocities. However, no significant mean biases were found for peak velocity. This was caused by the increased variability, as shown by the wider limits of agreements in both exercises (Figure 1 and Figure 2). Although Feuerbacher, et al. [9] did not investigate the peak velocity in relation to the T-Force and the LOA, they showed that with low loads of 30% of 1-RM, the LOA was much wider (0.17) than with higher loads (0.05–0.07). As the peak velocities in the present study are pretty similar to those at 30% in Feuerbacher, et al. [9] with almost the same LOAs (0.12–0.15); this could be an indication that at higher velocities, Vmaxpro has more variability.

The measurements with Speed4lifts in our study varied very much compared with T-Force. The ICCs varied from 0 to 0.99 and CVs over 10.3%, especially in peak velocities during bench presses; at most loads, this reliability was low (Figure 1 and Figure 2). Some of this low reliability is explainable by the possible outliers. However, after double checking, these measurements could not be deleted as inaccurate measurements. Furthermore, it does seem that Speed4lifts measures significantly lower velocities than T-Force, especially with higher velocities as observed with the low loads of 45 and 55% (Figure 3 and Figure 4) and the positive regression lines in the Bland–Altman plots for squats. Earlier research by Martínez-Cava, et al. [12] found that Speed4lifts showed high reliability at lower velocities but tended to show greater errors as the velocity increased, which was similar to what was found in the present study. In addition, the LOAs also differed very much between the exercises and peak and mean propulsive velocities; so were the LOAs of peak velocity for bench presses and mean propulsive velocity for squats, resp., 0.11 and 0.107 m/s, and they were comparable with the LOA range of GymAware and Musclelab at peak velocity (Figure 1 and Figure 2). However, the LOAs were much wider for the peak velocity in squats (0.20 m/s) and mean propulsive velocity in bench presses (0.147 m/s), which indicates large inaccuracies in measurements compared with T-Force (Figure 1 and Figure 2). These large differences were not observed in the study by Martínez-Cava, et al. [12], which is probably due to how the squats and bench presses were performed. In the present study, free weights were used, while Martínez-Cava, et al. [12] used a Smith machine for the exercises and thereby limited the lifting movement.

Lastly, PUSH band only measured mean velocity in both exercises and showed reliable CV and ICCs in squats, except at 1-RM, where the ICCs and CVs in bench press were not reliable according to the criteria with almost all loads (Table 2). Furthermore, velocities during both exercises were significantly higher than measured with T-Force (Figure 1 and Figure 2), especially in squats, where almost all loads with significantly higher velocities were measured with PUSH band (Figure 3). In addition, wide limits of agreements (bench press: 0.195 m/s, squat: 0.122 m/s) were observed, indicating inconsistent accuracy. Callaghan, et al. [7] found that PUSH band had acceptable reliability at certain loads but exhibited significant variability at higher loads and velocities, consistent with our findings. The findings are further supported by Jovanovic and Jukic [30] who, despite within-unit reliability, observed differences in agreement between GymAware and PUSH band, suggesting discrepancies can occur when comparing between them.

The observed differences in reliability and accuracy between the devices in the current study may be attributed to variations in the sampling rates of the different sensors being utilized (accelerometer vs. linear encoder), along with differences in data processing techniques. Devices such as linear encoders (GymAware, Musclelab, and Speed4lifts) typically use differentiation, measuring displacement directly and calculating velocity. In contrast, devices using integration, like accelerometer-based tools, calculate velocity from acceleration data, which can introduce cumulative errors and lead to greater variability while also being sensitive to measurement placement.

### 4.2. Actual vs. Estimated 1-RM Comparison

Our results indicate significant variability in the accuracy of 1-RM estimations among the different devices. When using mean propulsive velocity for estimation, only the Speed4lift and Musclelab devices showed significant underestimations of 1-RM in both the bench press and squat exercises. Specifically, Speed4lift underestimated 1-RM by 6.0 kg (5.8%) in squats and 7.9 kg (9.2%) in bench presses, while Musclelab underestimated 1-RM by 4.3 kg (4.8%) in bench presses (Table 3 and Table 4). These findings suggest that while some devices can closely estimate actual 1-RM on average, others tend to significantly underestimate it. Earlier research by Bosquet, et al. [10] observed high correlations between estimated and actual 1-RM using the Musclelab linear encoder but noted an average underestimation of 1-RM by 5.4 kg, which is similar to what is observed in the present study. The T-Force device, considered a gold standard, showed narrower limits of agreement and lower mean biases, reinforcing its reliability in estimating 1-RM. Furthermore, earlier research has highlighted the variability in the accuracy and reliability of commercially available velocity measurement devices. For example, Martínez-Cava, et al. [12] found that the T-Force device was preferable for identifying technical errors, with Speed4lifts and 3D motion capture systems serving as viable alternatives for medium to high loads. Moreover, Loturco, et al. [22] demonstrated the high precision of the bar velocity approach in predicting the maximum dynamic strength in bench press, supporting our observation that while certain devices are reliable at moderate loads, their accuracy diminishes at higher intensities. Similarly, Hughes, et al. [31] also emphasized caution with regards to the accuracy of the load–velocity relationship in predicting 1-RM.

### 4.3. Limitations

This study had limitations which must be addressed. Firstly, due to resources, a relatively small sample size from both sexes participated in the study. Yet, this small sample size mainly has influence upon the comparison between actual with predicted 1-RM loads. Although, for the comparison between the different equipment, all measurements with all loads were used, which resulted in 84 data points to compare. Nonetheless, due to including both sexes, a broader range of loads and velocities in the squat and bench press exercises were used, which made it possible to investigate the different measuring tools over a larger span. Additionally, this study only included two distinct exercises: bench presses and squats, which may limit the generalizability of the findings when performing other exercises.

## 5. Conclusions

This study assessed the reliability of various velocity measurements and compared actual 1-RM in the squat and bench press with the estimated value. GymAware revealed the highest reliability along with minimal bias, while Musclelab and Vmaxpro showed moderate reliability with some variability at higher loads. Speed4lifts and Push band indicated greater variability, specifically at higher intensities. Furthermore, in relation to the second aim of the study, significant discrepancies were found between actual and estimated 1-RM values, with Speed4lifts and Musclelab notably underestimating 1-RM. These findings underscore the importance of selecting reliable tools for accurate velocity-based training and load prescription.

## Figures and Tables

**Figure 1 sensors-24-07422-f001:**
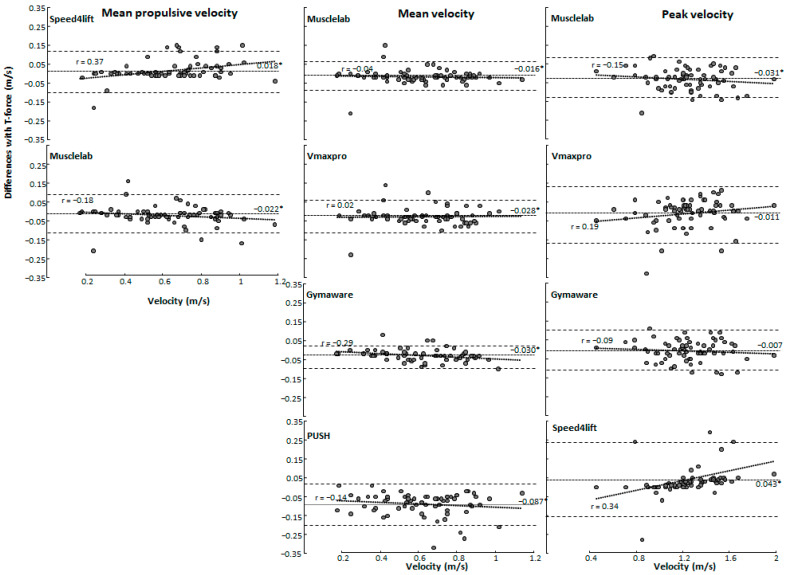
Bland–Altman plot with the mean bias and limits of agreement for mean propulsive velocity, mean velocity, and peak velocity between T-Force and the other velocity measurement devices in the squat. A negative value means that T-Force measures lower than the comparable measuring tool. * indicates a significant bias with T-Force on a *p* < 0.05 level.

**Figure 2 sensors-24-07422-f002:**
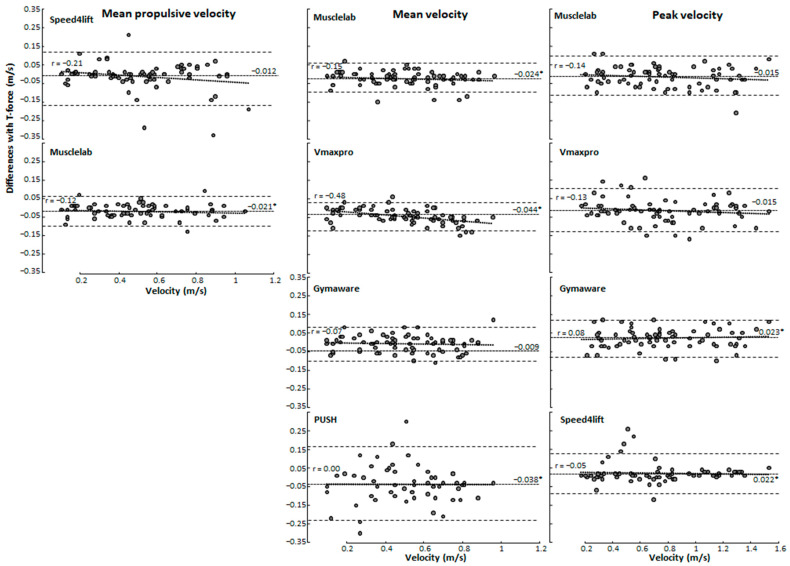
Bland–Altman plot with the mean bias and limits of agreement for mean propulsive velocity, mean velocity, and peak velocity between T-Force and the other velocity measurement devices in the bench press. A negative value means that T-Force measures lower than the comparable measuring tool. * indicates a significant bias with T-Force on a *p* < 0.05 level.

**Figure 3 sensors-24-07422-f003:**
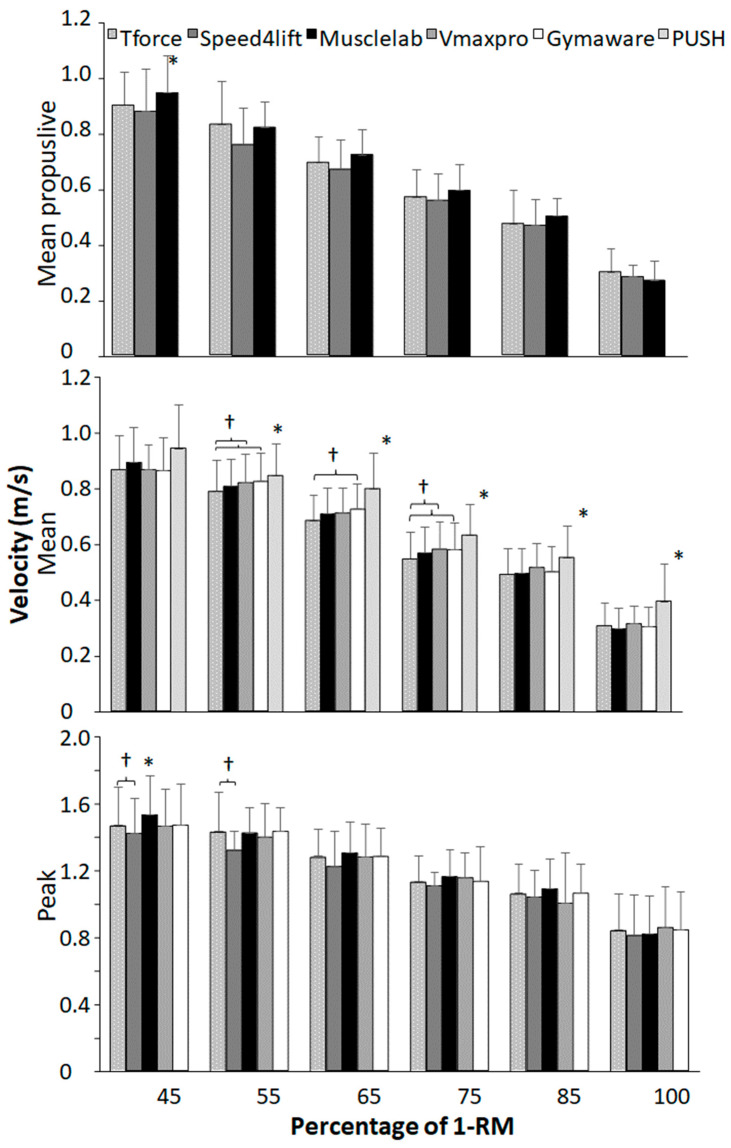
Average mean propulsive, mean, and peak (±SD) velocities per load in squats per measuring tool. * indicates a significant difference with all other measuring tools on a *p* < 0.05 level. † indicates a significant difference between these two measuring tools on a *p* < 0.05 level.

**Figure 4 sensors-24-07422-f004:**
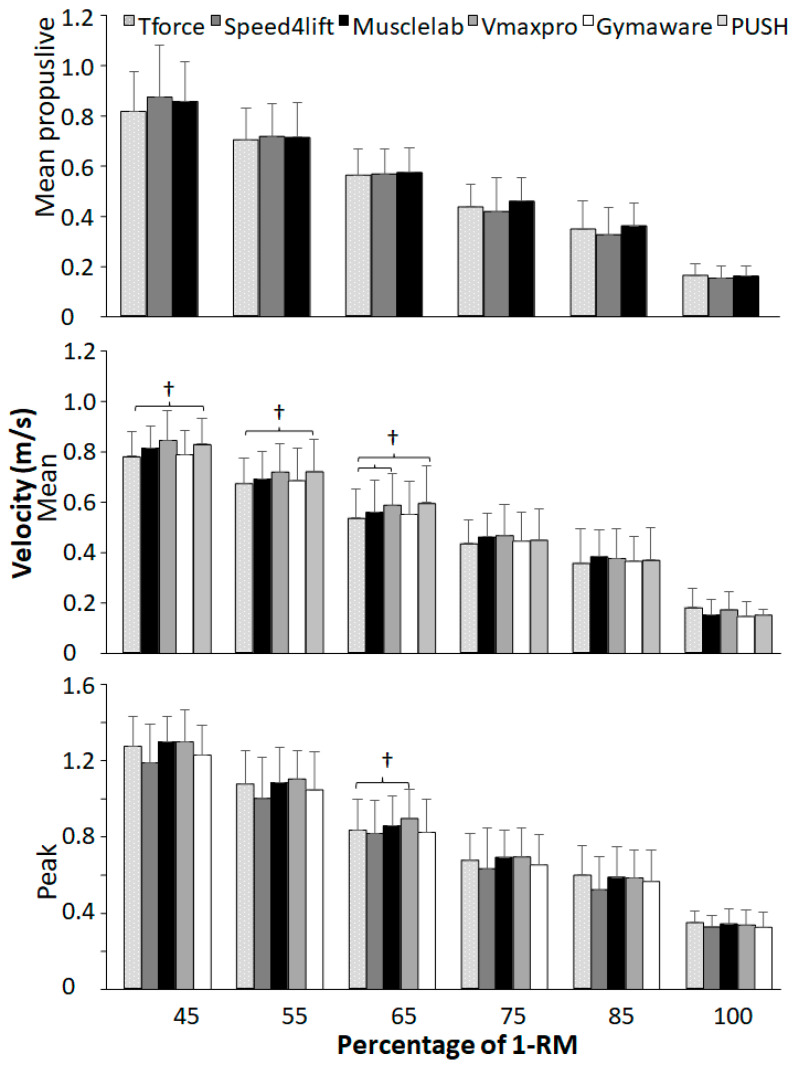
Average mean propulsive, mean, and peak velocities per load in bench presses per measuring tool. † indicates a significant difference between these two measuring tools on a *p* < 0.05 level.

**Table 1 sensors-24-07422-t001:** Coefficient of variation and intraclass correlation with 95% confidence intervals for mean propulsive velocity, mean velocity, and peak velocity between T-Force and the other velocity devices in the back squat.

Device	Load (%1-RM)	CV MPV	ICC (95% CI) MPV	CV MV	ICC (95% CI) MV	CV PV	ICC (95% CI) PV
Musclelab	45	3.7	0.88 (0.66–0.96)	1.3	0.99 (0.98–0.99)	2.1	0.98 (0.94–0.99)
	55	3.3	0.92 (0.77–0.98)	2.3	0.98 (0.93–0.99)	2.8	0.93 (0.79–0.98)
	65	4.2	0.88 (0.67–0.96)	2.0	0.98 (0.93–0.99)	3.3	0.94 (0.82–0.98)
	75	4.5	0.91 (0.74–0.97)	3.3	0.98 (0.93–0.99)	1.9	0.98 (0.94–0.99)
	85	9.4	0.80 (0.48–0.93)	9.2	0.97 (0.92–0.99)	5.2	0.89 (0.70–0.97)
	100	13.2	0.73 (0.34–0.91)	12.5	0.92 (0.75–0.98)	12.3	0.79 (0.45–0.93)
Speed4lifts	45	3.9	0.91 (0.75–0.97)	-	-	1.3	0.99 (0.98–0.99)
	55	4.6	0.83 (0.56–0.95)	-	-	7.1	0.46 (−0.09–0.80)
	65	3.3	0.93 (0.8–0.98)	-	-	6.6	0.80 (0.48–0.94)
	75	6.4	0.83 (0.56–0.95)	-	-	5.4	0.76 (0.37–0.92)
	85	12.9	0.63 (0.16–0.87)	-	-	15.3	0.00 (−0.52–0.53)
	100	7.3	0.94 (0.81–0.98)	-	-	7.8	0.92 (0.77–0.98)
Vmaxpro	45	-	-	8.2	0.97 (0.92–0.99)	2.3	0.98 (0.93–0.99)
	55	-	-	2.6	0.96 (0.87–0.99)	3.4	0.94 (0.81–0.98)
	65	-	-	4.6	0.88 (0.65–0.96)	4.8	0.89 (0.69–0.97)
	75	-	-	3.9	0.98 (0.94–0.99)	3.6	0.93 (0.78–0.97)
	85	-	-	10.4	0.89 (0.67–0.96)	2.8	0.97 (0.91–0.99)
	100	-	-	12.5	0.71 (0.29–0.90)	15.9	0.69 (0.25–0.89)
GymAware	45	-	-	13.2	0.96 (0.87–0.99)	2.6	0.97 (0.91–0.99)
	55	-	-	2.4	0.97 (0.90–0.99)	3.0	0.94 (0.81–0.98)
	65	-	-	3.4	0.93 (0.79–0.98)	3.5	0.94 (0.80–0.98)
	75	-	-	4.1	0.97 (0.92–0.99)	8.6	0.72 (0.31–0.90)
	85	-	-	3.5	0.97 (0.89–0.99)	3.1	0.96 (0.89–0.99)
	100	-	-	8.7	0.91 (0.75–0.97)	5.8	0.96 (0.87–0.99)
PUSH	45	-	-	6.3	0.80 (0.48–0.94)	-	-
	55	-	-	2.2	0.98 (0.93–0.99)	-	-
	65	-	-	5.3	0.87 (0.64–0.96)	-	-
	75	-	-	8.7	0.92 (0.75–0.97)	-	-
	85	-	-	5.3	0.93 (0.79–0.98)	-	-
	100	-	-	11.6	0.80 (0.47–0.94)	-	-

CV: Coefficient of variation, MPV: mean propulsive velocity, ICC: intraclass correlation, CI: confidence intervals, MV: mean velocity, PV: peak velocity.

**Table 2 sensors-24-07422-t002:** Coefficient of variation and intraclass correlation with 95% confidence intervals for mean propulsive velocity, mean velocity, and peak velocity between T-Force and the other velocity devices in the bench press.

Device	Load (% 1-RM)	CV MPV	ICC (95% CI) MPV	CV MV	ICC (95% CI) MV	CV PV	ICC (95% CI) PV
Musclelab	45	4.6	0.93 (0.81–0.98)	3.9	0.87 (0.66–0.95)	4.1	0.87 (0.64–0.96)
	55	5.6	0.93 (0.81–0.98)	2.7	0.96 (0.89–0.99)	3.7	0.95 (0.84–0.98)
	65	3.6	0.96 (0.88–0.99)	6.0	0.91 (0.75–0.97)	4.1	0.96 (0.89–0.99)
	75	4.0	0.96 (0.89–0.99)	3.4	0.97 (0.91–0.99)	4.6	0.95 (0.85–0.98)
	85	7.8	0.94 (0.83–0.98)	10.8	0.87 (0.64–0.96)	5.3	0.97 (0.90–0.99)
	100	15.2	0.81 (0.52–0.93)	16.1	0.65 (0.2–0.87)	12.8	0.74 (0.36–0.91)
Speed4lifts	45	9.7	0.79 (0.48–0.92)			11.7	0.36 (−0.19–0.74)
	55	9.7	0.73 (0.36–0.90)	-	-	14.1	0.41 (−0.13–0.77)
	65	4.5	0.95 (0.85–0.98)	-	-	2.9	0.95 (0.84–0.98)
	75	8.2	0.90 (0.74–0.97)	-	-	10.8	0.84 (0.58–0.95)
	85	6.4	0.96 (0.88–0.99)	-	-	19.7	0.60 (0.12–0.85)
	100	10.3	0.89 (0.7–0.96)	-	-	10.3	0.82 (0.52–0.94)
Vmaxpro	45	-	-	3.8	0.91 (0.74–0.97)	4.1	0.89 (0.68–0.96)
	55	-	-	3.5	0.94 (0.82–0.98)	2.8	0.96 (0.89–0.99)
	65	-	-	3.8	0.97 (0.90–0.99)	7.1	0.91 (0.74–0.97)
	75	-	-	4.7	0.95 (0.87–0.99)	6.6	0.90 (0.72–0.97)
	85	-	-	7.0	0.95 (0.84–0.98)	5.4	0.97 (0.89–0.99)
	100	-	-	15.4	0.78 (0.45–0.92)	11.6	0.78 (0.43–0.92)
GymAware	45	-	-	4.8	0.82 (0.53–0.94)	3.2	0.93 (0.80–0.98)
	55	-	-	5.2	0.89 (0.69–0.96)	4.0	0.95 (0.84–0.98)
	65	-	-	6.1	0.92 (0.76–0.97)	3.9	0.94 (0.81–0.98)
	75	-	-	5.6	0.93 (0.80–0.98)	6.0	0.93 (0.79–0.98)
	85	-	-	8.9	0.91 (0.75–0.97)	5.5	0.96 (0.88–0.99)
	100	-	-	18.4	0.60 (0.13–0.85)	8.6	0.79 (0.27–0.89)
PUSH	45	-	-	4.6	0.85 (0.59–0.95)	-	-
	55	-	-	15.1	0.82 (0.53–0.94)	-	-
	65	-	-	9.8	0.82 (0.54–0.94)	-	-
	75	-	-	11.3	0.76 (0.41–0.92)	-	-
	85	-	-	35.1	0.00 (−0.51–0.51)	-	-
	100	-	-	26.9	0.53 (0.03–0.82)	-	-

CV: Coefficient of variation, MPV: mean propulsive velocity, ICC: intraclass correlation, CI: ≤confidence intervals, MV: mean velocity, PV: peak velocity.

**Table 3 sensors-24-07422-t003:** Estimated 1-RM in bench presses and squats based upon load–velocity relationship of mean propulsive and mean velocity as measured by the different equipment and actual 1-RM.

	Squat (kg)		Bench Press (kg)	
Calculation Method	Mean Propulsive Velocity	Mean Velocity	Mean Propulsive Velocity	Mean Velocity
*Actual 1-RM*	*116.5 ± 38.8*		*90.8 ± 31.9*	
T-Force	113.5 ± 39.4	114.7 ± 39.5	87.4 ± 32.6	87.3 ± 32.4
Speed4lift	110.5 ± 39.4 *		82.9 ± 32.0 *	
Musclelab	114.1 ± 41.3	115.4 ± 41.3	86.5 ± 30.3 *	87.7 ± 31.8
Vmaxpro	-	116.9 ± 42.8	-	88.0 ± 33.2
GymAware	-	115.8 ± 41.4	-	88.1 ± 34.2
PUSH 2.0	-	120.1 ± 46.5	-	89.7 ± 30.2

* indicates a significant difference with the actual measured 1-RM load on a *p* < 0.05 level.

**Table 4 sensors-24-07422-t004:** Absolute and relative difference between estimated and actual 1-RM in bench presses and squats based upon the load–velocity relationship of mean propulsive (MPV) and mean velocity measured by the different tools.

	Squat				Bench Press			
	Absolute (kg)		Percentage (%)		Absolute (kg)		Percentage (%)	
Calculation Method	MPV	Mean Velocity	MPV	Mean Velocity	MPV	Mean Velocity	MPV	Mean Velocity
T-Force	−3.0 ± 7.8	−1.8 ± 8.4	−2.6 ± 7.5	−1.4 ± 8.4	−3.4 ± 5.9	−3.5 ± 9.0	−4.5 ± 6.8	−4.4 ± 9.2
Speed4lift	−6.0 ± 5.0 *		−5.8 ± 6.0 *		−7.9 ± 7.1 *		−9.2 ± 7.1 *	
Musclelab	−2.4 ± 4.9	−1.1 ± 4.9	−3.0 ± 5.3	−1.8 ± 5.0	−4.3 ± 5.8 *	−3.1 ± 6.6	−4.8 ± 5.6 *	−3.7 ± 7.0
Vmaxpro	-	0.4 ± 7.5	-	−0.3 ± 7.4	-	−2.8 ± 7.4	-	−3.7 ± 7.7
GymAware	-	−0.7 ± 4.0	-	−1.5 ± 4.5	-	−2.6 ± 9.0	-	−3.8 ± 9.4
PUSH 2.0	-	3.6 ± 11.4	-	1.5 ± 10.5	-	−6.2 ± 9.5	-	−5.8 ± 9.2

* indicates a significant difference with the actual measured 1-RM load on a *p* < 0.05 level.

## Data Availability

The raw data supporting the conclusion of this article will be made available by the authors without undue reservation.

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
