# Peer review of "Comparison of Velocity and Estimated One Repetition Maximum Measured with Different Measuring Tools in Bench Presses and Squats"

_sensors, 2024, doi:10.3390/s24237422_

Round 1

Reviewer 1 Report

Comments and Suggestions for Authors

Thank you very much for reviewing your interesting study. The research question is important and the relevance is outlined in parts in the introduction. 

The methods section must be improved, while the results part would benefit from a more detailed statistical evaluation, although it is well-performed in general.

The discussion seems to be more a replication of previous study results (Which were, in parts, inappropriate in their evaluation). Therefore, I would recommend to take a more balanced perspective there, discuss the strength and weaknesses of previous results and the practical relevance. 

My point-to-point comments are listed below, I hope you find them helpful for revising the manuscript. Looking forward to reading it published, as I will probably refer to this work in the future several times, if aspects listed below will be addressed. 

Lines 27 – 36: The rationale is not clear to me. You state: “Training load or proximity-to-failure may be a crucial training variable for different physiological adaptations (i.e., strength and muscle hypertrophy) following resistance training [3]. Hence, velocity-based resistance training has become popular as this could measure concentric velocity and therefore prescribe training load [4].“ The red line can be improved where there is the benefit of velocity based training. Where there is the relevance of performing velocity maximal strength estimation and why it should provide any benefits compared to common 1RM testing and load control.

Lines 45 – 49: It should be also acknowledged that evaluating correlation coefficients is, per se, no valid method to assess validity and/or accuracy.

Lines 48 – 52: The sentences read not smoothly, please consider rephrasing.

Lines 52 – 60: “The 52 researchers concluded that the T-Force device was the preferrable option to identify po-tential technical errors among monitoring technologies. In addition, they concluded that Speed4Lifts and the 3D motion capture system were fine alternatives to T-Force when  training with medium and high loads, whereas My Lift showed substantial measurement  errors. Also, the Speed4lift has been observed to be the most reliable device among seven  commercially available devices to measure velocity [5], whereas the authors concluded  that IMUs were less accurate velocity measures compared to linear velocity/position transducers, camera-based systems, and smartphone applications [5].“ Can you please provide the reader with further information? Why were these sensors classified the most accurate and reliable? Which statistical key indices led the authors to conclude that the Speed4Lifts and 3D MoCap were finde alternatives? How was agreement determined and where are thresholds that underscore this classification? This also applies to the subsequent classification for the 1m/s and <0.65m/s velocities.

Lines 65 following: Feuerbacher reported, on the one hand LoAs and performed a BA analysis, however, the authors unfortunately forgot to appropriately interpret the results. It seems that almost all differences plotted in the BA analysis indicate sufficient reliability and the reference values for the span of LoAs were not provided. If they were, the authors would have to recognize that the random error was unappropriately high.

Providing Ris additionally relatively useless to state validity. (as mentioned above).

Lines 76 following: The rationale, that no study investigated the reliability is a very weak point. A content related rationale should be provided why reliability in this case is of special interest, why a comparison between sensors would be relevant and, in special, why the propulsive velocity as an uncommon parameter is of additional benefit that deserves special consideration.

Lines 83/84: Why did you chose 5 velocities and why these 5? I think for instance the EnodePro makes an (invalid) estimation of the 1RM after just one trial. For a linear regression, just 2 trials were needed, while the manuals handbook says that 3 trials (at least) should be performed. Can you please clarify why you used 5? Especially the high loads (75/85%)? This could be derived in the introduction in a more content related way.

Methods

14 participants is a comparatively small sample size. What do you think about the statistical power? You did just test the submaximal loads on one testing occasion. How stable are these velocities and how familiar were the participants with the testing protocol? Is there a rationale why no habituation session was conducted and why no inter-day reliability was provided for the submaximal velocities? If the objective of this work is to assess reliability in general, I would assume that I find inter- and intraday reliability quantifications (no just for the 1RM).

Procedures

Does it really make sense from your perspective to perform 5-10 repetitions in just one set and then (at least some participants) directly start with 85% 1RM or 75% 1RM? Especially in the squat, this raises diverse concerns regarding safety and validity of the trials. 

Where do I find the description of the exercises? Did I miss them? Was it a deep squat, parallel squat? Which bench press and squatting technique was performed and how was it standardized ? 2 years of resistance training does not necessarily means experience with the squat, which is a highly complex and techniqally highly demanding exercise with several aspects which must be considered. Which guidelines were followed?

Statistics

You write you performed a BA Analysis WITH an ANOVA. How and for what? Please clarify

Furthermore, in a simulation study was shown that an ICC of 0.7 was accompanied by a random error (secondary variance) of around 20%. How do you think about a CV of 10% is still sufficient and an ICC of 0.7 would indicate sufficient reliability? Although there are references available, I think there should be a content based classification. If I perform two times in a row the exact same measurement, I think, as a scientist, you should not be satisfied by a measurement error of 10-20%, should you?

Correlation statistics were not derived in the introduction, as you aimed to investigate if the devices measure the same value, not how they are related. Furthermore, although I understand the intention of performing a one-way ANOVA, it depends on your hypothesis whether this is appropriate. Wouldn’t you test for separated hypotheses (so agreement between enode and T-Force) instead of general differences between the testing results ? It might be of neglectable relevance, but in case of separated analyses, the accumulation of type one error seems negotiable and this procedure could be performed.

Results

As mentioned above, if we decrease the threshold for “acceptable” reliability further, all reliability values are sufficient. Why not using shrout and fleiss? I suggest a content based reliability classification

For the reliability analysis I would suggest accounting for random errors (MAE;MAPE) and to quantify the systematic error. You report that there is one in the agreement between the devices, however, how large is the error. It is complicated to read it off the BA plots. Furthermore, the exact same parameters apply to the reliability determination, as we would expect in a reliable measure that we would have agreement between the first and second value. Therefore, I suggest to add these information to check whether there was a learning effect between first and second trial, or how large the secondary variance was. 

As mentioned, please add quantification of the random error and classify the LoAs based on the mean differences (a velocity difference of 0.1m/s might be neglectable with a mean of 100m/s, however it must be interpreted differently if there is a range of ±0.1 surrounding a mean of 0.001 (yes these are extreme examples, however, they might make clear the intended meaning). 

Discussion

Lines 302 – 306: it reads as you aimed to check the agreement between the T-Force as the baseline and then to compare it with other devices. However, why was this performed? Why is the T-Force considered the most valid/reliable tool and, more importantly, how were the reliability values compared to the other devices? Especially as the MuscleLab or the EnodePro seem to better predict the 1RM in the mean? To me it would be of interest how the individual scattering was performed. Even though the predictability in the mean might fit, what is with individuals in which the 1RM is overestimated with about 30% (https://www.frontiersin.org/journals/physiology/articles/10.3389/fphys.2024.1435103/full). Especially if evaluating the practical relevance, we must consider individual scattering before recommending devices for practice, as it seems dangerous to exclusively focus on means and SDs. For this aspect, it is without relevance if we are talking about the construct of reliability or validity – precision must be given in both cases.

Lines  346-348: First: would you classify muscle lab with this ICC range really as reliable? Did you quantify different error sources? (https://pubmed.ncbi.nlm.nih.gov/10907753/,https://pubmed.ncbi.nlm.nih.gov/9820922/, https://pubmed.ncbi.nlm.nih.gov/17613641/ ) I would recommend to consider these references for further analyses steps maybe and to focus on classifying the sensors in the light of their practical applications: Screening the daily performance of individuals. 

Furthermore, you write “with some variability”…. Which type of statistical quantification is some variability? Please try to be more specific.

Lines 353: What is an acceptable mean bias and how was it classified and on which scale are the stated LoAs acceptable? Acceptable for what?

Lines 358 – 361: “Previous research by Feuerbacher, et al. [9], who assessed the validity of Vmaxpro found high validity (r=0.935 and 0.968 when compared to Vicon and T-Force) and moderate to high reliability (ICC=0.66–0.94), suggesting reliability and acceptable performance of Vmaxpro in measuring velocities. However, Feuerbacher, et al. [9] observed an overestimation of mean velocities. “

About 5 problems in just such a small paragraph. If velocities are overestimated, how can the reliability and accuracy be accepted? Validity (again) is not quantified via correlation coefficients). I suggest to handle previous literature with substantial deficits more carefully and critically.

Lines 362 folloiwng: Now you quantify 0.65 -0.96 as moderate reliability, while the same range was classified moderate to large previously. This is confusing to me.

Lines 372- 374: When are LoAs wide and when acceptable for you? This lacks objective quantifications.

In the reliability part I miss a critical and comprehensive discussion on the problems and limitations of this method arising from the stated reliability indices. It looks like a reoplication of previous study results only (which can be a part of this section). However, the paper would greatly benefit from a more controversial standpoint by balancing the listed aspects with some opposing arguments and considering more relevant statistical key values.

This point of discussion also applies to the following section (Lines 390 following).

Limitations: So why did you not recruite more participants? It is not a very complicated study design with large measurement effort. 

Sex differences were not included to your research question so not investigating them is no limitation of this study. You also did not investigate the influence of the hair cut on the evaluated parameters. Why did you not list this in the limitations as well. Please focus on limitations of the study design (the sample size is one important aspect here). 

If you investigate device reliability and validity I am not aware why it is a limitation to include trained athletes? It is not on the athlete but on the device. You can also just move the bar up and down and investigate the reliability and the validity of the device (which was even more appropriate as you could move the bar with a pre-determined velocity). I strongly recommend to list only real limitations instead of listing aspects that did actually not bias answered your research question.

Author Response

We want to thank the reviewers for their time to review the manuscript. We think that we have now answered to all the comments of the reviewers and think that the manuscript now is suitable for publication. All changes are marked in red in the manuscript.

Reviewer 1

Thank you very much for reviewing your interesting study. The research question is important and the relevance is outlined in parts in the introduction.

Thank you

The methods section must be improved, while the results part would benefit from a more detailed statistical evaluation, although it is well-performed in general.

We have now changed parts of the methods part and updated the statistical evaluation

The discussion seems to be more a replication of previous study results (Which were, in parts, inappropriate in their evaluation). Therefore, I would recommend to take a more balanced perspective there, discuss the strength and weaknesses of previous results and the practical relevance.

We have rewritten parts of the discussion and think that the discussion is much better now.

My point-to-point comments are listed below, I hope you find them helpful for revising the manuscript. Looking forward to reading it published, as I will probably refer to this work in the future several times, if aspects listed below will be addressed.

Thank you for your review. We will try to answer to all your comments point-to-point.

Lines 27 – 36: The rationale is not clear to me. You state: “Training load or proximity-to-failure may be a crucial training variable for different physiological adaptations (i.e., strength and muscle hypertrophy) following resistance training [3]. Hence, velocity-based resistance training has become popular as this could measure concentric velocity and therefore prescribe training load [4].“ The red line can be improved where there is the benefit of velocity based training. Where there is the relevance of performing velocity maximal strength estimation and why it should provide any benefits compared to common 1RM testing and load control.

We have rewritten this section a bit and now added a sentence after request from the reviewer:

Unlike traditional one-repetition maximum (1-RM) testing, which may require the lifter to perform repetitive maximal lifts to track 1RM progression, estimating the load-velocity relationship through one single 1-RM test could enable continuous tracking of the load-velocity relationship, and thus continuous tracking of changes in the lifters performance (4).

Lines 45 – 49: It should be also acknowledged that evaluating correlation coefficients is, per se, no valid method to assess validity and/or accuracy.

We agree that correlation coefficients or no valid method for validity. Here we wanted to show that there is a very large correlation between actual and estimated 1-RM. But we already state in the next sentence that Musclelab significantly underestimated 1-RM with around 5kg (bias). We have included significantly and bias, to make it clear that this is a real difference.

Lines 48 – 52: The sentences read not smoothly, please consider rephrasing.

We have changed the sentence in: Thereby, according to the authors this estimation is not accurate enough to prescribe resistance training intensities based upon this 1-RM estimations predicted by that software [10].

Lines 52 – 60: “The researchers concluded that the T-Force device was the preferrable option to identify potential technical errors among monitoring technologies. In addition, they concluded that Speed4Lifts and the 3D motion capture system were fine alternatives to T-Force when  training with medium and high loads, whereas My Lift showed substantial measurement  errors. Also, the Speed4lift has been observed to be the most reliable device among seven  commercially available devices to measure velocity [5], whereas the authors concluded  that IMUs were less accurate velocity measures compared to linear velocity/position transducers, camera-based systems, and smartphone applications [5].“ Can you please provide the reader with further information? Why were these sensors classified the most accurate and reliable? Which statistical key indices led the authors to conclude that the Speed4Lifts and 3D MoCap were fine alternatives? How was agreement determined and where are thresholds that underscore this classification? This also applies to the subsequent classification for the 1m/s and <0.65m/s velocities.

We have now changed this part with more detailed information after request from the reviewer.

Acceptable reliability criteria were set to the smallest detectable change of < 0.07 m/s. The authors concluded that the T-Force device was the preferrable option to identify potential technical errors among monitoring technologies due to the high observed intra-device reliability across the load-velocity spectrum as they observed the smallest detectable change to be 0.01-0.02 m/s. In addition, they concluded that Speed4Lifts and the 3D motion capture system were fine alternatives to T-Force when training with medium and high loads as the smallest detectable change were 0.01-0.05 m/s, and 0.02-0.04 m/s for the Speed4lifts and the 3D motion capture system, respectively. In another study, both the PUSH Band 2.0 and Speed4lifts were checked for test-retest reliability against a 3D motion capture system in both squat and bench press The authors found that both devices were reliable with slow (<0.65 m/s) and moderate (0.65-1.00 m/s), but not fast (>1.0 m/s) velocities as they observed intraclass correlations of >0.7 for both devices during the slow and moderate velocities, and intraclass correlations of <0.64 for the fast barbell loads.

Lines 65 following: Feuerbacher reported, on the one hand LoAs and performed a BA analysis, however, the authors unfortunately forgot to appropriately interpret the results. It seems that almost all differences plotted in the BA analysis indicate sufficient reliability and the reference values for the span of LoAs were not provided. If they were, the authors would have to recognize that the random error was unappropriately high.

We have checked the reference again and agree fully with the reviewer. We have changed the text now in: Vmaxpro had high correlations (R2 > 0.93) when compared to both the motion capture system and T-force. However, Vmaxpro overestimated mean velocities with an average of 0.06 m/s compared to the other velocity measurements with large limits of agreements. Furthermore, the observed systematic bias with the other two measurement tools in-creased with higher mean velocities  indicating a heteroscedasticity when measuring mean velocity with different loads.

Providing R2 is additionally relatively useless to state validity. (as mentioned above).

We have changed this now in correlations in the text.

Lines 76 following: The rationale, that no study investigated the reliability is a very weak point. A content related rationale should be provided why reliability in this case is of special interest, why a comparison between sensors would be relevant and, in special, why the propulsive velocity as an uncommon parameter is of additional benefit that deserves special consideration.

We agree wit the reviewer that that is not a good rationale. We have included a rationale now. As many training studies use one of these sensors to measure velocity at different load and it could be of importance to know if these measurement with different loads are com-parable between these measurement tools. This makes it easier for trainers and researchers to compare findings performed with these different measurement tools with each other and if they need to adjust results when comparing these measurement tools.

About the propulsive velocity, this is a parameter, which is also used by several measurement tools and takes away the error at low loads (deceleration phase). That is why this one is also included. This is also mentioned in the text now: … as these are the parameters that are often used to estimate 1-RM and thereby not known how accurate the use of these parameters are.

Lines 83/84: Why did you chose 5 velocities and why these 5? I think for instance the EnodePro makes an (invalid) estimation of the 1RM after just one trial. For a linear regression, just 2 trials were needed, while the manuals handbook says that 3 trials (at least) should be performed. Can you please clarify why you used 5? Especially the high loads (75/85%)? This could be derived in the introduction in a more content related way.

We used these 5 loads as it represents the whole range at which strength training with this type of equipment is used. The main purpose was to investigate the velocity measurement between the T-force with the other measurement tools and what the effect of different loads was. We agree that for a linear regression 2 or 3 loads are needed for establishing a 1-RM prediction. But as said before the first purpose was to compare velocity output at different loads of the different measurement tools as these are very comment and trainers need to know if these are similar when using different types of tools over these loads. This is now mentioned in the introduction.

Methods

14 participants is a comparatively small sample size. What do you think about the statistical power? You did just test the submaximal loads on one testing occasion. How stable are these velocities and how familiar were the participants with the testing protocol? Is there a rationale why no habituation session was conducted and why no inter-day reliability was provided for the submaximal velocities? If the objective of this work is to assess reliability in general, I would assume that I find inter- and intraday reliability quantifications (no just for the 1RM).

The main purpose was to compare the different measurement tools with the T-force and see how comparable they are. Since we have used 6 different loads the number of used data points are much more than just 14 as shown in the Blant-Altman plots. Only for comparing the estimated with the actual 1-RM the number of subjects could be low. However, based upon the sample size by Gpower only 12 subjects were necessary. This is now written in the methods part:

To determine adequate sample size between estimated and actual 1-RM measurements between the six measurement tools, an a priori power analysis was calculated using G*Power (version 3.1.9.2, University of Kiel, Kiel, Germany) using the f test family (ANOVA repeated measures, within factors, f = 0.8, α = 0.05 and power of 0.80) and based upon the findings of Bosquet, et al. [10] and Fitas, et al. [18]. The analysis revealed that a total sample size of n = 12 would be sufficient to find significant and medium-sized effects of estimated 1-RMs of measurement tools with actual 1-RM loads. 

As mentioned by the reviewer did we test everything in just one session as many of the measurement tools already have tested their test-retest reliability in other studies. We did not also want to test this again as this would increase the size of the manuscript even more. No habituation session was performed as all subjects perform these exercise on a regular basis with these different range of loads (depends on the part of the training season). So they are used with this. As said before no inter-day reliability was performed as the main purpose was to compare the measurement tools with each other. It is not to expected that this will change between days as equipment does not know when the test is performed and we were not interested in testing the test-retest reliability of each measurement tool, but only the numbers between each other.

Procedures

Does it really make sense from your perspective to perform 5-10 repetitions in just one set and then (at least some participants) directly start with 85% 1RM or 75% 1RM? Especially in the squat, this raises diverse concerns regarding safety and validity of the trials.

The subjects in this study, are strength trained and normally also start after just one set of easy weights with 85%. Thus, we have noticed now problems with this randomized set up. Furthermore, we wanted to avoid fatigue. As we already have 5 different load before conducting 1-RM we wanted to avoid to much warm-up sets. As said before, these subjects were used to start heavy after just one warm-up set.

Where do I find the description of the exercises? Did I miss them? Was it a deep squat, parallel squat? Which bench press and squatting technique was performed and how was it standardized ? 2 years of resistance training does not necessarily means experience with the squat, which is a highly complex and technically highly demanding exercise with several aspects which must be considered. Which guidelines were followed?

We have included more information about the background of the subjects and we have also included more information about the description of the exercises as these were not specified as pointed out by the reviewer.

These texts are included to the manuscript:

… which they train squats and bench press. Inclusion criteria specified that subjects had to manage a squat equivalent to 1.5 × body mass (men) and 1 × body mass (women) [18], following to the technique requirements established by the International Powerlifting Federation. In bench press men had to lift in 1-RM at least 1.2 × body mass and women 0.9 × body mass [19] to be sure that they have an appropriate lifting technique in both exercises. Additionally, subjects had to declare absence of any injury or illness, which could hinder maximum effort.

The subjects performed the bench press and squat according to the rules and regulations set by the International Powerlifting Federation. Thus, the depth through the end of the descent phase during the squat was measured and standardized with the depth requirement from International Powerlifting Federation and marked with a horizontal band behind the subject that needed to touched to start the ascent. In bench press, the requirement for a full stop on the chest was removed; they were allowed to touch and press, but no bounce of the barbell was allowed. Both stance width, barbell placement, and external rotation of the feet during squats and grip width during bench press were self-selected by the subjects but standardized for each individual throughout all loads. The descent movement velocity was self-selected, but the ascent phase had to pe performed with maximal intensity in each attempt.

Statistics

You write you performed a BA Analysis WITH an ANOVA. How and for what? Please clarify

We agree with the reviewer that this should be specified. We have now included that it was on the bias between T-force with the other measurement tools and between the estimated 1-RM with actual 1-RM loads. We hope this is clear now.

Furthermore, in a simulation study was shown that an ICC of 0.7 was accompanied by a random error (secondary variance) of around 20%. How do you think about a CV of 10% is still sufficient and an ICC of 0.7 would indicate sufficient reliability? Although there are references available, I think there should be a content based classification. If I perform two times in a row the exact same measurement, I think, as a scientist, you should not be satisfied by a measurement error of 10-20%, should you?

We used the standard interpretation of sufficient reliability as stated in earlier studies, which are an ICC>0.7 and CV<10%. So we are not satisfied with an CV of more than 10%. So we agree that that is not reliable enough.

Correlation statistics were not derived in the introduction, as you aimed to investigate if the devices measure the same value, not how they are related. Furthermore, although I understand the intention of performing a one-way ANOVA, it depends on your hypothesis whether this is appropriate. Wouldn’t you test for separated hypotheses (so agreement between enode and T-Force) instead of general differences between the testing results ? It might be of neglectable relevance, but in case of separated analyses, the accumulation of type one error seems negotiable and this procedure could be performed.

As we wanted to see how other measurement tools relate to the T-force (gold standard in our opinion as shown in previous studies). That is why we performed one-way ANOVAs to investigate if the bias between them are significant or not and if different percentages of loads have an influence upon this evt. bias at low, moderate and high velocities. That is also why we tested a correlation over the different loads to investigate of there is a tendency of bias increase ot lower or higher velocity between the devices.

To avoid accumulation of type one error we have chosen for just comparing with the T-force as we think it reflects the golden standard of linear encoders as shown in previous studies. This also results in less comparisons between the different measurement tools to avoid this increased change of a type 1 error.

Results

As mentioned above, if we decrease the threshold for “acceptable” reliability further, all reliability values are sufficient. Why not using shrout and fleiss? I suggest a content based reliability classification

The authors appreciate the reviewers perspective. However, the guidelines from Shrout and Fleiss are rules of thumb, whereby interpretation and usage will vary by the field of research. It is the authors opinion that a more conservative approach is justified by the need for precision in measurements tested in validation studies. Therefore, the threshold for acceptable reliability is based upon similar research validating velocity measurements with different loads (García-Ramos, et al (2018).

For the reliability analysis I would suggest accounting for random errors (MAE;MAPE) and to quantify the systematic error. You report that there is one in the agreement between the devices, however, how large is the error. It is complicated to read it off the BA plots. Furthermore, the exact same parameters apply to the reliability determination, as we would expect in a reliable measure that we would have agreement between the first and second value. Therefore, I suggest to add these information to check whether there was a learning effect between first and second trial, or how large the secondary variance was.

In our opinion, in the present study we compare the velocities of the T-force with the other measurement tools in reliability and validity. In the BA plots we have shown over all loads if there is a systematic bias and also the LOAs. We have marked with * if the bias was significant with the T-force for this parameter. We also tested in those figures the heteroscedasticity. We did not include the numbers of the LOAs in the figures as we think that it will be too much numbers, which makes the figures not easy to read anymore. It was not our purpose to compare reliability within each system, which we could do if we investigate also rep 1 and 2 of the different loads. When we also would do these analysis this would increase the total number of analysis even more and would make the story less readable for the readers. Reviewer 2 already suggests less information in the results part. We hope that the reviewer understands our point of view. The main purpose is the comparison between the T-force on different loads with the other tools and the 1-RM estimations.

We see in the data that at the lower loads (45-65) the highest velocity was found in rep 2, while at higher loads it was the first rep. This is already shown in many previous studies with higher loads that the velocity decreases wit reps. That is why we used the maximal velocity at each load for further analysis.

As mentioned, please add quantification of the random error and classify the LoAs based on the mean differences (a velocity difference of 0.1m/s might be neglectable with a mean of 100m/s, however it must be interpreted differently if there is a range of ±0.1 surrounding a mean of 0.001 (yes these are extreme examples, however, they might make clear the intended meaning).

We think that we have shown this in the BA plots in figure 1 and 2 We have not written the exact numbers of the LOAs here as we think that it will make the figures less readable due to the overflow of information in the figures. The lines indicate already what the lower and upper levels are, which we think gives a good view over the errors over the different loads. We hope the reviewer agrees with us about this.

Discussion

Lines 302 – 306: it reads as you aimed to check the agreement between the T-Force as the baseline and then to compare it with other devices. However, why was this performed? Why is the T-Force considered the most valid/reliable tool and, more importantly, how were the reliability values compared to the other devices? Especially as the MuscleLab or the EnodePro seem to better predict the 1RM in the mean? To me it would be of interest how the individual scattering was performed. Even though the predictability in the mean might fit, what is with individuals in which the 1RM is overestimated with about 30% (https://www.frontiersin.org/journals/physiology/articles/10.3389/fphys.2024.1435103/full). Especially if evaluating the practical relevance, we must consider individual scattering before recommending devices for practice, as it seems dangerous to exclusively focus on means and SDs. For this aspect, it is without relevance if we are talking about the construct of reliability or validity – precision must be given in both cases.

We have decided to use the T-force as gold standard as indicated in the introduction that it is the most valid one compared to 3D measuring tools. After reading several studies that have indicated this. Also, these studies have shown that T-fore very accurately estimates the 1-RM correctly.

In the study you refer to 2 repetitions are compared. However, earlier studies have already looked at variability and reliability when using several repetitions. That was not the scope of our study and would make the results part even longer. We only wanted to compare the values between T-force and other measurement tools. Furthermore, in practice, most of the times the maximal measured velocity or the average of 2-3 attempts per load are used to estimate 1-RM. We have decided to use only the maximal velocity in each load as mentioned in the methods part.

Lines  346-348: First: would you classify muscle lab with this ICC range really as reliable? Did you quantify different error sources? (https://pubmed.ncbi.nlm.nih.gov/10907753/,https://pubmed.ncbi.nlm.nih.gov/9820922/, https://pubmed.ncbi.nlm.nih.gov/17613641/ ) I would recommend to consider these references for further analyses steps maybe and to focus on classifying the sensors in the light of their practical applications: Screening the daily performance of individuals.

We fully agree with the reviewer that with this ICC range Musclelab is not qualified to be reliable enough. The lower ICCs with high CV show that at 100% the measurements are not accurate enough. This is now also mentioned in the text now. The whole part is changed now.

Furthermore, you write “with some variability”…. Which type of statistical quantification is some variability? Please try to be more specific.

We have deleted this part to avoid confusion and rewritten this part.

Lines 353: What is an acceptable mean bias and how was it classified and on which scale are the stated LoAs acceptable? Acceptable for what?

When the mean bias is not significantly different from 0 then that is acceptable. About the LoAs, we don’t know what acceptable is and to avoid that we have changed this in small ranges to avoid confusion.

Lines 358 – 361: “Previous research by Feuerbacher, et al. [9], who assessed the validity of Vmaxpro found high validity (r=0.935 and 0.968 when compared to Vicon and T-Force) and moderate to high reliability (ICC=0.66–0.94), suggesting reliability and acceptable performance of Vmaxpro in measuring velocities. However, Feuerbacher, et al. [9] observed an overestimation of mean velocities.”

About 5 problems in just such a small paragraph. If velocities are overestimated, how can the reliability and accuracy be accepted? Validity (again) is not quantified via correlation coefficients). I suggest to handle previous literature with substantial deficits more carefully and critically.

We fully agree with the reviewer about all these points on this. We have deleted most of this and rewritten the whole paragraph on the Vmaxpro in: The Vmaxpro demonstrated similar reliabilities as musclelab with also a too low ICC  of 0.69 with peak velocity during squats, and too low CVs with maximal loads (Table 1 and 2). Furthermore, also Vmaxpro measures higher mean velocities compared to the T-force as shown by signifcant mean biases (Figure 1 and 2) and higher velocities at loads of 55 and 75% in squats (Figure 3) and at 65% in bench press (Figure 4). This was in line with Feuerbacher, et al. [1] who also observed an overestimation of mean velocities. However, no signficant mean biases were found for peak velocity. This was caused by the increased variability as shown by the wider limits of agreements in both exercises (Figure 1 and 2). Although Feuerbacher, Jacobs, Dragutinovic, Goldmann, Cheng and Schumann [1] did not investigate the peak velocity in relation to the T-force and the LOA, they showed that with low loads of 30% of 1-RM the LOA was much wider (0.17) than with higher loads (0.05-0.07). As the peak velocities in the present study are pretty similar to those at 30% in Feuerbacher, Jacobs, Dragutinovic, Goldmann, Cheng and Schumann [1] with the almost same LOAs (0.12-0.15) this could be an indication that at higher velocities the Vmaxpro has more variability.  

Lines 362 following: Now you quantify 0.65 -0.96 as moderate reliability, while the same range was classified moderate to large previously. This is confusing to me.

Again, this is very confusing. We have rewritten this part also in: The measurements with the Speed4lifts in our study varied very much compared with the T-force. The ICCs varied from 0 to 0.99 and CVs over 10.3%, especially in peak velocities during bench press at most loads this reliability was low (Figure 1 and 2). Some of this low reliability is explainable by the possible outliers. However, after double checking these measurements could not be deleted as inaccurate measurements. Furthermore, does it seem that Speed4lifts measures significantly lower velocities than the T-force, especially with higher velocities as observed that the low loads of 45 and 55% (Figure 3 and 4) and the positive regression lines in the Bland-Altman plots for squats. Earlier research by Martínez-Cava, et al. [12], found that Speed4lifts showed high reliability at lower velocities but tended to show greater errors as the velocity increased, which was similar to what was found in the present study. In addition, the LOAs differed also very much between the exercises and peak and mean propulsive velocities. So were the LOA of peak velocity at bench press and mean propulsive velocity in squats resp. 0.11  and 0.107 m/s and comparable with the LOA range of Gymaware and Musclelab at peak velocity (Figure 1 and 2). However, the LOAs were much wider for the peak velocity in squats (0.20 m/s) and mean propulsive velocity in bench press (0.147 m/s), which indicates large inaccuracies in measurements compared to the T-force (Figure 1 and 2). These large differences were not observed in the study by Martínez-Cava, et al. [12], which is probably due to how the squats and bench press were performed. In the present study free weights were used, while Martínez-Cava, et al. [12] used a Smith machine for the exercises and thereby limited the lifting movement.

Lines 372- 374: When are LoAs wide and when acceptable for you? This lacks objective quantifications.

We don’t know what is acceptable. We now changed it in just mentioning that they are wider, which is an indication that accuracy is less between the devices. If the reviewer has an objective quantification of what is acceptable, please inform us about this.

In the reliability part I miss a critical and comprehensive discussion on the problems and limitations of this method arising from the stated reliability indices. It looks like a replication of previous study results only (which can be a part of this section). However, the paper would greatly benefit from a more controversial standpoint by balancing the listed aspects with some opposing arguments and considering more relevant statistical key values.

We have rewritten most of the discussion with the comments of the reviewer and think that the discussion is much better now.

This point of discussion also applies to the following section (Lines 390 following).

We think that this part of the discussion is pretty clear now.

Limitations: So why did you not recruite more participants? It is not a very complicated study design with large measurement effort.

When we did the measurements it was vacation here (bad excuse), which limited the access to subjects. Furthermore, we needed subjects of a certain strength level, which also limits access to athletes. The place where tested is just a small town, which also limits the number of athletes. However, for the main purpose of the study of comparing the different measurement tools with the T-force, this number of athletes is enough as shown in earlier similar studies and by the use of the different loads during the test. We hope that the reviewer now is enough informed about our limited access of subjects.

Sex differences were not included to your research question so not investigating them is no limitation of this study. You also did not investigate the influence of the hair cut on the evaluated parameters. Why did you not list this in the limitations as well. Please focus on limitations of the study design (the sample size is one important aspect here).

We fully agree with the reviewer and deleted that sentence. We have rewritten the limitations part in: Yet, this small sample size has mainly influence upon the comparison between actual with predicted 1-RM loads. Though, for the comparison between the different equipment all measurements with all loads were used, which resulted in 84 data points to compare. Nonetheless, …

If you investigate device reliability and validity I am not aware why it is a limitation to include trained athletes? It is not on the athlete but on the device. You can also just move the bar up and down and investigate the reliability and the validity of the device (which was even more appropriate as you could move the bar with a pre-determined velocity). I strongly recommend to list only real limitations instead of listing aspects that did actually not bias answered your research question.

Again, we agree with the reviewer. We have changed it now in: Additionally, the study only included two distinct exercises: bench press and squats, which may limit the generalizability of the findings when performing other exercises.

Reviewer 2 Report

Comments and Suggestions for Authors

Comparison of velocity and estimated 1-RM measured with different measuring tools in bench press and squats REVIEW

This study aimed to compare barbell velocities at different intensities and estimated 1-RM with actual 1-RM measured with different measuring tools in bench press and squats.

The usage use and comparison of numerous technical solutions for measuring muscle function, is study key strength.

Small subject sample is main weakness of this study. The study has only 14 participants. That is why the generalisation of such findings is questionable.

Abstract is informative but I advise authors to give their own recommendation for testing device of choice based on study results.

Introduction section lacks theoretical explanation of force velocity relationship (this is mandatory for publishing this paper).  

Also, open question is whether it makes sense to study muscle strength with a two gender sample and whether in that case the final result should be partialized for body mass.  This question should be addressed in introduction to make present study justified.

I advise authors to give in more details how and why they made study sample (power analyses, inclusion-exclusion criteria and number of cases, sampling method...).  Also, we need more information on level of competition at raining of participants.

Since we have sample of two genders it is not clear are study design, data collecting and processing method adequate for obtaining answer to research question. This could be judged only after additional argumentation.

The results of this study gives clear and detailed answer to research question. Never the less presented results are to detailed and redundant so I advise authors to make condensation of them and point out clearly key findings. 

Results are discussed appropriately, and conclusion is supported by the both results and discussion. All of this opens the space for technological improvement in muscle function assessment in practise.

The references are appropriate.

I have not additional comments.

Author Response

We want to thank the reviewers for their time to review the manuscript. We think that we have now answered to all the comments of the reviewers and think that the manuscript now is suitable for publication. All changes are marked in red in the manuscript.

Reviewer 2

Comparison of velocity and estimated 1-RM measured with different measuring tools in bench press and squats REVIEW

This study aimed to compare barbell velocities at different intensities and estimated 1-RM with actual 1-RM measured with different measuring tools in bench press and squats.

The usage use and comparison of numerous technical solutions for measuring muscle function, is study key strength.

Thank you. That is also the main purpose of the present study.

Small subject sample is main weakness of this study. The study has only 14 participants. That is why the generalization of such findings is questionable.

The main purpose was to compare the different measuring tools with each other. For this 14 subjects with 6 different loads should be enough as we have shown in the study. However, for comparing calculated 1-RM with actual 1-RM the number of subjects could be a limiting factor. This is now specified more in the limitations part.

Abstract is informative but I advise authors to give their own recommendation for testing device of choice based on study results.

We would be very interested in including our recommendation to the abstract. However, we are limited by the maximal word count of 200 words. Since we already have 198 words and don’t see where we can delete parts of the abstract to make space for a recommendation, we can’t include this to the abstract. Sorry.

Introduction section lacks theoretical explanation of force velocity relationship (this is mandatory for publishing this paper).

Firstly it is not a force velocity relationship we have investigated, but a load-velocity relationship. This is now changed in the manuscript. We have now also included what the this load-velocity relationship is so the readers are informed about this now. This is included to the text: This is the relationship between the load lifted and the corresponding barbell velocity at which it is lifted during different loads on which, by linear regression analysis, a relationship is found that is often used to predict 1-RM loads in different exercises [15]. 

Also, open question is whether it makes sense to study muscle strength with a two gender sample and whether in that case the final result should be partialized for body mass.  This question should be addressed in introduction to make present study justified.

The main purpose of the present study was to compare different measuring tools with the findings of the T-force (gold standard), it does not matter if we used men or women as velocity at different loads  are compared between the different equipment. Sex does not play a role in this. Only perhaps in establishing calculated 1-RM a different 1-RM can be found as previous studies found another load-velocity relationship between men and women. However, as the main purpose were the measurements were between the different measuring tools and it was a within subject design, difference in maximal strength between the sexes was of no importance.

I advise authors to give in more details how and why they made study sample (power analyses, inclusion-exclusion criteria and number of cases, sampling method...).  Also, we need more information on level of competition at training of participants.

We have included the inclusion criteria to be in the study now under the subject. This is mentioned in the text: in … which they train squats and bench press. Inclusion criteria specified that subjects had to manage a squat equivalent to 1.5 × body mass (men) and 1 × body mass (women) [18], following to the technique requirements established by the International Powerlifting Federation. In bench press men had to lift in 1-RM at least 1.2 × body mass and women 0.9 × body mass [19] to be sure that they have an appropriate lifting technique in both exercises. Additionally, subjects had to declare absence of any injury or illness, which could hinder maximum effort.

We have now also added the part of sample size calculations in the text. For the Bland-Alman plots this is not necessary as there are 84 data points used. For the comparison of estimated with actual 1-RM we have used Gpower and needed only 12 subjects. This is mentioned in the text now.

To determine adequate sample size between estimated and actual 1-RM measurements between the six measurement tools, an a priori power analysis was calculated using G*Power (version 3.1.9.2, University of Kiel, Kiel, Germany) using the f test family (ANOVA repeated measures, within factors, f = 0.8, α = 0.05 and power of 0.80) and based upon the findings of Bosquet, et al. [10] and Fitas, et al. [18]. The analysis revealed that a total sample size of n = 12 would be sufficient to find significant and medium-sized effects of estimated 1-RMs of measurement tools with actual 1-RM loads. 

 Since we have sample of two genders it is not clear are study design, data collecting and processing method adequate for obtaining answer to research question. This could be judged only after additional argumentation.

As the main purpose was to compare the reliability and accuracy between the different measuring tools with the T-force findings, it did not matter that subjects from different sexes were used as it is a comparison between equipment and not between sexes. The only difference is that women lifted less then men and perhaps have another type of load-velocity relationship, this would not result in different outputs between equipment. Furthermore, velocity was measured at different percentages of 1-RM which is similar for both sexes and thereby also does not have an influence upon the findings. As earlier studies have found different load-velocity relationships between sexes, this could result in different calculated 1-RM loads between sexes. However, as the load-velocity relationships were based upon each individual and compared with their actual load, again no sexes difference would occur here in our opinion.

The results of this study gives clear and detailed answer to research question. Never the less presented results are to detailed and redundant so I advise authors to make condensation of them and point out clearly key findings.

Here we kindly disagree with the reviewer as we have highlighted the main findings in the abstract and the discussion. We think all findings are of importance. Thus, we would like to have the results chapter as it is. We hope that this is okay for the reviewer.

Results are discussed appropriately, and conclusion is supported by the both results and discussion. All of this opens the space for technological improvement in muscle function assessment in practise.

Thank you

 The references are appropriate.

Thank you

 I have not additional comments.

Round 2

Reviewer 1 Report

Comments and Suggestions for Authors

No further comments.

Reviewer 2 Report

Comments and Suggestions for Authors

Thanx for your cooperation.